ecology, evolution, genomics

wildlife disease, *Batrachochytrium dendrobatidis*, genetic epidemiology, population genetics, amphibians

**Author for correspondence:**
Andrew P. Rothstein
e-mail: andrew.rothstein@berkeley.edu

# Divergent regional evolutionary histories of a devastating global amphibian pathogen

Andrew P. Rothstein[1,2], Allison Q. Byrne[1,2,3], Roland A. Knapp[4,5], Cheryl J. Briggs[5,6], Jamie Voyles[7], Corinne L. Richards-Zawacki[8] and Erica Bree Rosenblum[1,2]

[1]Department of Environmental Science, Policy, and Management and [2]Museum of Vertebrate Zoology, University of California Berkeley, Berkeley, CA, USA
[3]Center for Conservation Genomics, Smithsonian Conservation Biology Institute, National Zoological Park, Washington, DC, USA
[4]Sierra Nevada Aquatic Research Laboratory, University of California, Mammoth Lakes, CA, USA
[5]Earth Research Institute and [6]Department of Ecology, Evolution, and Marine Biology, University of California, Santa Barbara, CA, USA
[7]Department of Biology, University of Nevada, Reno, NV, USA
[8]Department of Biological Sciences, University of Pittsburgh, Pittsburgh, PA, USA

APR, 0000-0002-1329-8702

Emerging infectious diseases are a pressing threat to global biological diversity. Increased incidence and severity of novel pathogens underscores the need for methodological advances to understand pathogen emergence and spread. Here, we use genetic epidemiology to test, and challenge, key hypotheses about a devastating zoonotic disease impacting amphibians globally. Using an amplicon-based sequencing method and non-invasive samples we retrospectively explore the history of the fungal pathogen *Batrachochytrium dendrobatidis* (*Bd*) in two emblematic amphibian systems: the Sierra Nevada of California and Central Panama. The hypothesis in both regions is the hypervirulent Global Panzootic Lineage of *Bd* (*Bd*GPL) was recently introduced and spread rapidly in a wave-like pattern. Our data challenge this hypothesis by demonstrating similar epizootic signatures can have radically different underlying evolutionary histories. In Central Panama, our genetic data confirm a recent and rapid pathogen spread. However, *Bd*GPL in the Sierra Nevada has remarkable spatial structuring, high genetic diversity and a relatively older history inferred from time-dated phylogenies. Thus, this deadly pathogen lineage may have a longer history in some regions than assumed, providing insights into its origin and spread. Overall, our results highlight the importance of integrating observed wildlife die-offs with genetic data to more accurately reconstruct pathogen outbreaks.

## 1. Introduction

Globalization has contributed to a surge in the incidence, severity and spread of emerging infectious diseases (e.g. [1,2]). Emerging diseases of wildlife are particularly important to global biological diversity as they can cause devastating population declines and exacerbate other threats such as habitat loss, overharvesting, invasive species and climate change [3–6]. Recent advances in the study of disease emergence and spread integrate epidemiological and genetic data to test theoretical predictions about the ecological history of the pathogen given the underlying evolutionary signal [7,8]. However, most applications of this approach have been for quickly evolving pathogens (i.e. RNA viruses) and those that directly impact human health. There have been a handful of studies applying methodological advances in genetic epidemiology to emerging

wildlife diseases (see recent reviews [9,10]), but such frameworks are still largely underutilized.

Amphibians are declining worldwide [11,12]. One of the major drivers of amphibian declines is the global spread of the disease chytridiomycosis, caused by the fungal pathogen *Batrachochytrium dendrobatidis* (*Bd*) [13]. *Bd* infects the keratinized skin cells of susceptible host species, disrupts vital amphibian skin functions and can cause mortality [14]. In some cases, *Bd* infections can spread quickly across individuals, populations, and species leading to pizootic outbreaks and population and community collapses [15,16]. Since the earliest observations of *Bd*-related die-offs in the late 1990s, *Bd* has emerged as a global threat to amphibian biodiversity and now impacts amphibians on every continent where they are present [12].

*Bd* has a complex evolutionary history with multiple lineages found in different parts of the world. Phylogenetically, *Bd* is characterized by several early branching lineages endemic to different regions (*Bd*CAPE, *Bd*ASIA1, *Bd*Brazil/ASIA2 and *Bd*ASIA3) and one more recently derived hyper-virulent panzootic lineage (*Bd*GPL). *Bd*GPL has been linked to declines of amphibian communities around the world and is the only *Bd* lineage with a truly global distribution [6,17,18]. Whole-genome data have been important for revealing the dynamics of *Bd*GPL spread [6,19]. *Bd*GPL typically exhibits little phylogenetic or spatial genetic structure (with the exception of two subclades *Bd*GPL-1 and *Bd*GPL-2) [20,21], suggesting that this lineage spread rapidly around the world [18,19]. Moreover, compared to other *Bd* lineages, *Bd*GPL genomes have fewer pairwise genetic differences among them and highly variable genetic diversity values [6]. Observations of minimal pairwise genetic differences are consistent with rapid *Bd*GPL spatial radiation, and variability in genetic diversity suggests episodes of population size fluctuation. However, we still lack a connection between our understanding of *Bd* evolutionary history at a global scale and regional *Bd* emergence and spread.

Two of the most emblematic *Bd*GPL-related declines occurred in the montane amphibian communities of the Sierra Nevada of California and Central Panama. In the Sierra Nevada of California, mountain yellow-legged frogs (*Rana sierrae/muscosa*), were historically one of the most abundant vertebrates [22]. Over the last century, these frogs vanished from more than 90% of their historic range, and *Bd* (along with invasive fish) was a significant factor in their decline [23]. Available information suggests that *Bd* has been spreading across the Sierra Nevada since at least the 1960s [24,25] and has caused epizootics and subsequent extirpations in hundreds of populations [16,26,27]. Some populations that experienced *Bd*-related declines are beginning to rebound, but remaining naive populations are still at risk for *Bd* epizootics [28]. Similarly, in Central America, amphibian population declines were first observed in the late 1980s [29,30]. As *Bd* spread southeast into Central Panama starting in the early 2000s [15], many susceptible amphibian host species declined—or even disappeared completely—across the region [15,31,32]. Although some species seem to be recovering [32], *Bd*-related declines have fundamentally reshaped these tropical communities [31,33,34].

From an epizological perspective, amphibian declines in the Sierra Nevada and Central Panama appear quite similar. In both regions, initial detection of *Bd* was followed by devastating outbreaks and host mortality. Patterns of decline in both the Sierra Nevada and Central Panama also appear to provide evidence of a 'wave'-like spread of *Bd* across the landscape [16,35]. Pathogen prevalence and population decline data in both systems suggest that new infections appear in a predictable spatial direction and that *Bd* outbreaks move a predictable distance each year [15,16,35]. Coupled with a global phylogenetic view of *Bd*, the prevailing hypothesis suggests that *Bd*GPL is a recent invasive pathogen in these two regions [36]. However, epizoological data based on observed outbreaks and host outcomes may or may not reflect the true history of *Bd* arrival and spread. The Sierra Nevada and Central Panama differ dramatically in climate, habitat, and amphibian community composition. Ecological and environmental factors impact the physiological limits of *Bd*, transmission dynamics across the landscape and host immunity; extensive evidence suggests these factors influence prevalence and host disease outcomes [20,37–39]. Stark differences between regional environments likely contribute to different *Bd* dynamics in the Sierra Nevada and Panama, making their apparent similarities in disease outcomes all the more intriguing. Although it is often assumed that *Bd* arrived recently and spread in a wave-like fashion in both regions, it is possible that different evolutionary histories of *Bd* underlie these observed patterns.

Molecular data can reveal nuances of a pathogen's history that cannot be obtained by field observations alone. Genetic and genomic approaches have previously been used to investigate the evolutionary history of *Bd* at regional and global scales [6,18,19,40]. However, most studies of the evolutionary history of *Bd* in emblematic systems like the Sierra Nevada and Central Panama have relied on a small number of *Bd* isolates for any one region. Live and pure *Bd* cultures have been the source of high-quality DNA for genomic sequencing (e.g. [6,19,41]) but are inherently challenging to obtain, isolate and maintain. Low sample sizes and poor spatial coverage has made it difficult to test fine-scale hypotheses about *Bd* emergence and spread. However, advances in sequencing technology now allow for leveraging fine-scale sampling of frog skin swabs, previously used to determine *Bd* presence/absence and load, to robustly characterize *Bd* genotypes across relevant spatial scales [42]. Thus, we can now test whether patterns of *Bd* emergence that appear similar across systems result from shared underlying processes.

We used fine-scale genetic sampling to investigate assumptions about the history of *Bd*GPL in the Sierra Nevada and Central Panama. Using non-invasive skin swabs collected across similar spatial and temporal scales, we targeted hundreds of loci across the *Bd* genome to examine the hypothesis of recent *Bd* emergence and unidirectional epizootic spread in these two emblematic systems. Our work provides an in-depth understanding of pathogen evolutionary dynamics in natural systems and an example for how researchers should not expect infectious diseases to emerge and spread similarly across the globe.

## 2. Material and methods

### (a) Sampling and genotyping

We used skin swab DNA samples collected from the Sierra Nevada and Central Panama across similar timescales (2011–2017) and across equivalent spatial scales (approx. 130 km across Euclidean distance between furthest two sites) (figure 1a).

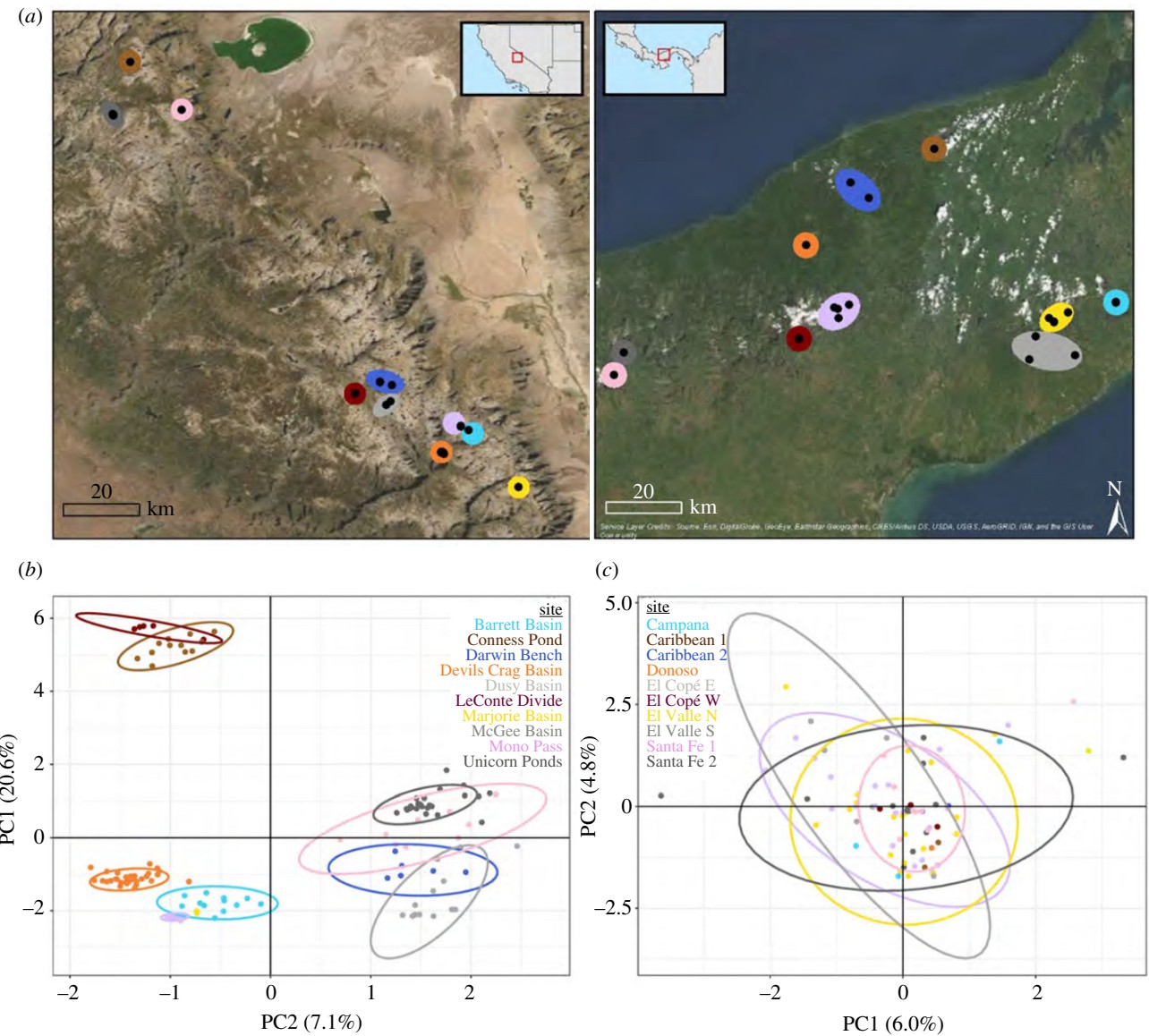

**Figure 1.** Study system map and principal component analysis of within region genotypes. (*a*) Map of sites sampled in the study in the Sierra Nevada and Central Panama. (*b*) PCA within Sierra Nevada samples, coloured by the major site. Samples cluster by site, suggesting strong genetic structuring across the Sierra Nevada. (*c*) PCA within Central Panama samples, coloured by the site. Compared to samples from the Sierra Nevada, Central Panama samples exhibit a dramatically different pattern, i.e. panmixis, despite a similar spatial and temporal scale of sampling. Colours in (*b*,*c*) correspond to geographical locations in (*a*). (Online version in colour.)

Sites are defined as collections of lakes and streams that cluster together geographically within a region. We sampled 10 sites from both the Sierra Nevada ($n_{samples} = 130$; $n_{species} = 2$) and Central Panama ($n_{samples} = 80$; $n_{species} = 17$). Sierra Nevada samples comprised skin swabs from two sister species of frogs (*R. sierrae/muscosa*) [23] and Central Panama samples comprised skin swabs from 16 different frog species. Additionally, we included 120 previously published sequenced samples from a global *Bd* dataset, comprised of samples across 59 frog species from six continents (see electronic supplementary material). The global samples were all previously positioned within the *Bd*GPL clade based on a comprehensive assessment of hundreds of *Bd* samples [17] and serve to add a global context to levels of genetic structure and diversity observed in the Sierra Nevada and Central Panama.

We sequenced 240 regions (each 150–200 bp long) of the *Bd* genome from the Sierra Nevada and Central Panama skin swab samples. We first conducted a pre-amplification step (which improves performance for amplicon sequencing) and then used these pre-amplified products in a microfluidic PCR approach (see electronic supplementary materials for DNA extraction, preparation and PCR conditions) [42]. Pre-amplified products were loaded into a Fluidigm Access Array IFC,

individually barcoded, then pooled for sequencing on ¼ of an Illumina MiSeq lane with $2 \times 300$ bp paired-end reads at the University of Idaho IBEST Genomics Resources Core. From raw sequence reads, we used the dbcAmplicons software (https://github.com/msettles/dbcAmplicons) to trim adapter, primer sequences and merged to continuous reads. We de-multiplexed and filtered sequences using the *reduce_amplicons.R* script within the dbcAmplicons repository into two sequence types: ambiguities and raw fastq for each sample. Ambiguities sequence files used IUPAC ambiguity codes to identify multiple alleles. Raw fastq files are all sequences for each sample. Ambiguity sequences were used for phylogenetic analyses and the fastq by the sample was used for alignment, variant calling and filtering VCF for downstream analyses. Specific parameters used for alignment, variant calling and variant filtering can be found in the electronic supplementary material. Post filtering our raw 4534 variants, we recovered 2268 variable sites across 235 amplicons.

## (b) Genetic diversity

We applied PCA to examine genetic clustering and structuring among all samples. We estimated PCs using *adegenet* [43] and

visualized in R (v. 3.6.1). We calculated summary diversity statistics using ANGSD [44]. For this analysis, diversity statistics were calculated based on genotype likelihoods, which is distinct from the variant calling approach by Freebayes described above. Given that sample sizes can impact diversity metrics, we randomly subsampled our the Sierra Nevada and global $Bd$GPL samples to equal the number of Central Panama samples ($n = 80$). Additionally, 49 amplicons (previously developed as Central Panama-specific markers) were removed from the filtered 235 amplicons, leaving 186 amplicons for diversity statistics. Using filtered BAMs from our variant calls, we generated a folded site frequency spectrum given an unknown ancestral state. After estimating the site frequency spectrum for each region, we calculated per-site Watterson's $\theta$ and $\pi$ for the Sierra Nevada (mean amplicon depth = 85.3; s.d. = 78.9; range = 7.0–464.8), Central Panama (mean amplicon depth = 207.8; s.d. = 206.2, range = 15.4–909.6) and global $Bd$GPL (mean amplicon depth = 323.7; s.d. = 383.1; range = 5.5–1342.5) samples. We tested for significant differences in mean Watterson's $\theta$ and $\pi$ and using analysis of variance followed by Tukey's HSD in R (v. 3.6.1), given that we had multiple pairwise comparisons of our global $Bd$GPL reference, Sierra Nevada and Central Panama samples.

## (c) Phylodynamics

Using ambiguity sequences by sample, we created a phylogeny including Sierra Nevada, Central Panama and our global $Bd$GPL reference panel. We removed amplicons that had no data and included samples that had least 20 amplicons. We trimmed loci that had greater than 5 bp difference between the minimum and maximum sequence length to control for improper alignments near large indels. A final list of 206 loci were individually aligned using the MUSCLE package in R (v. 3.6.1, [45]) and concatenated (28 688 bp in length). We also included previously published sequences of UM142 $Bd$Brazil as an outgroup [17]. Using this concatenated alignment, we built a phylogeny using IQ-Tree with 1000 ultrafast bootstrap replicates and chose the best model from AIC scores using 'model-finder' (GTR + F + I + G4) [46].

We assessed temporal signal in our phylogeny using TempEst (v. 1.5.3) and a date randomization test with TipDatingBeast (v. 1.1) (details available in electronic supplementary material, Methods) [47,48]. After confirming the temporal signal, we inferred time-measured phylogenies, with our concatenated alignment and recorded sampling years, using both BEAST2 [49] and Nextstrain [50]. We used time-measure parameters from a previously published whole-genome phylogenies for $Bd$ to ensure comparability [6]. Briefly, for BEAST2 we used a GTR substitution model with estimated mutation rates $7.29 \times 10^{-7}$ (lower; $3.41 \times 10^{-7}$, upper; $1.14 \times 10^{-6}$) and extended Bayesian skyline plot as demographic parameter [6]. Using this model, we ran a chain which drew samples every 3000 MCMC steps from a total of 575 000 000 steps, after a discarded burn-in of 57 500 000 steps. Convergence of distribution and effective sample size greater than 150 were checked through *Tracer* (v. 1.7.1) [51]. Our best-supported tree was estimated using maximum clade credibility through *TreeAnnotator* (v. 2.6) [49] and was visualized using *FigTree* (v. 1.4.4) (https://github.com/rambaut/figtree). Comparative methods for Nextstrain can be found in the electronic supplementary material.

It is important to note that we used BEAST2 and Nextstrain as analytical frameworks to compare patterns between the Sierra Nevada and Central Panama but not to infer exact introduction dates. Applying the same evolutionary models across two geographical regions provides a powerful comparative tool and allows us to infer *relative* evolutionary rates and introduction timings. However, we interpret specific dates with caution given that

patterns of $Bd$ genome evolution may violate a number of model assumptions (e.g. variation across the genome in recombination and mutation rates, variation in chromosomal copy numbers, the potential for both meiotic and mitotic recombination) [19,41], and because our sampling dates do not necessarily correspond to first introduction dates. Given that any violation of basic model assumptions would be shared across study regions, comparisons between the Sierra Nevada and Central Panama can be used to draw conclusions about the relative invasion history in these regions.

# 3. Results

## (a) *Bd* from the Sierra Nevada shows greater population structure than *Bd* from Central Panama

When comparing within regions, we found significant genetic clustering across the Sierra Nevada (figure 1b) but no genetic clustering across Central Panama (figure 1c). Samples collected from the same site in the Sierra Nevada clustered together, regardless of collection year or species (electronic supplementary material, figure S1). Starting with Unicorn Ponds at the north, samples generally follow a pattern of isolation by distance. LeConte Divide and Conness Pond are somewhat anomalous, however, because they overlap in PC space but are geographically separated by approximately 80 km (figure 1b). By contrast, Central Panama genotypes exhibited panmictic patterns, regardless of locality, collection year or species, indicating no genetic structuring across a similar spatio-temporal scale (figure 1c, electronic supplementary material, figure S2).

## (b) *Bd* from the Sierra Nevada shows greater variation and diversity than *Bd* from Central Panama

We confirmed that $Bd$ from the Sierra Nevada and Central Panama belong to the global $Bd$GPL lineage. Both Sierra Nevada and Central Panama samples clustered with a panel of global samples that were previously identified as $Bd$GPL [17]. Interestingly, the Sierra Nevada and Central Panama samples clustered separately from each other in PC space when compared to global $Bd$GPL samples (figure 2a). Additionally, we found that overall genetic diversity was significantly higher in the Sierra Nevada as compared to Central Panama [Tukey HSD, $p < 0.0001$] (figure 2b,c). Remarkably, we also found that Sierra Nevada $Bd$ samples have comparable and, in the case of Watterson's $\theta$, higher diversity than the set of global $Bd$GPL samples [Tukey HSD, $p < 0.0001$]. When comparing Central Panama and the Sierra Nevada using individual sites with similar samples sizes, we found that the majority of Sierra Nevada sites had higher mean diversity compared to Central Panama sites (both Watterson's $\theta$ and $\pi$) [Tukey HSD, $p < 0.001$], except in the lowest sample size pairing ($N = 5$) where El Valle S. had significantly higher mean diversity than LeConte Divide (electronic supplementary material, figure S3).

## (c) *Bd* in the Sierra Nevada is inferred to be older than *Bd* in Central Panama

Using a time-dated phylogenetic approach that included previously published global $Bd$GPL samples for reference [17], we found branches from Sierra Nevada samples were

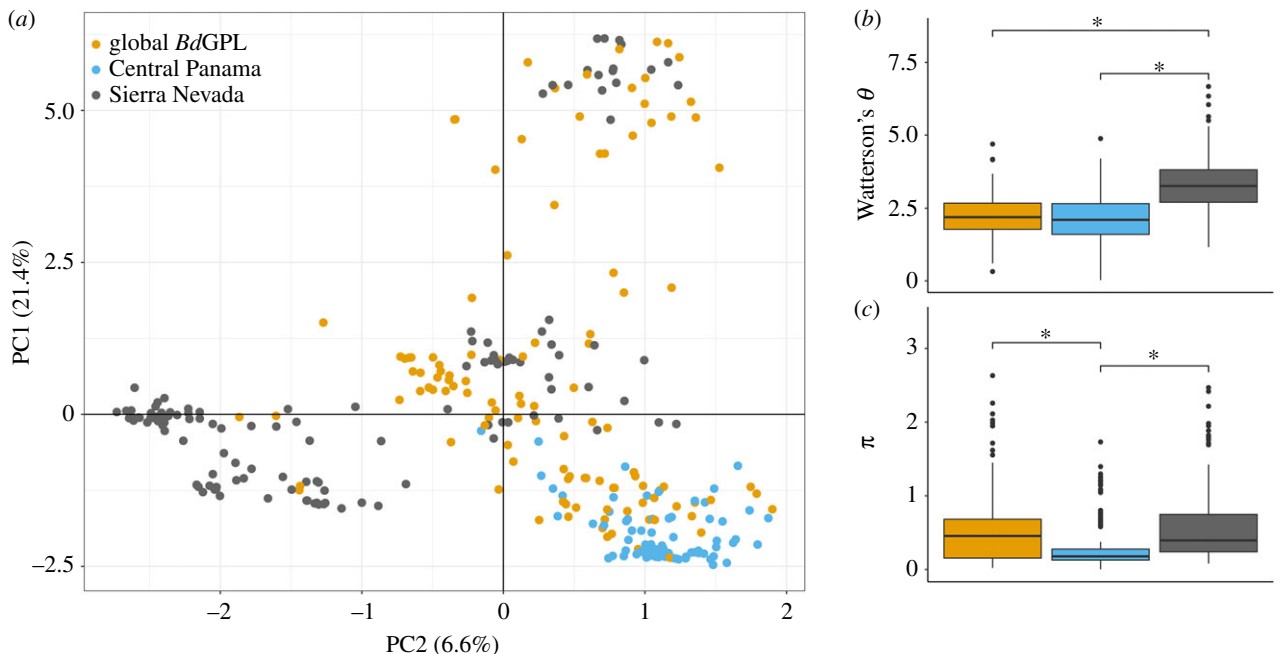

**Figure 2.** Genetic differentiation and diversity among the Sierra Nevada, Central Panama, and global *Bd*GPL samples. (*a*) PCA based on *Bd*GPL genotypes from the Sierra Nevada ($n = 130$), Central Panama ($n = 80$) and global reference panel ($n = 120$). Colours indicate samples from each region. The global reference panel included samples from dozens of frog species across all continents with *Bd*GPL. Samples from the Sierra Nevada and Central Panama are almost entirely separated in PC space with the Sierra Nevada samples showing greater genetic variation than Central Panama samples. (*b*) Distribution of mean genetic diversity (Watterson's $\theta$) for all variable sites based on region. Samples from the Sierra Nevada and global panels were randomly subsampled to match Central Panama sample size (all regions $n = 80$). Mean genetic diversity was significantly higher for Sierra Nevada samples compared to Central Panama samples and to the global *Bd*GPL panel [Tukey HSD, $p < 0.0001$]. (*c*) Distribution of mean nucleotide diversity ($\pi$) for all variable sites based on the region using the same samples as (*b*). Mean nucleotide diversity was significantly lower for Central Panama samples compared to Sierra Nevada samples and the global *Bd*GPL panel [Tukey HSD, both $p < 0.0001$]. Each box plot shows the median (horizontal line), first and third quartiles (bottom and top of box, 'hinges'), lowest and highest values within inter-quartile range of the lower and upper hinges (vertical lines) and outliers (points). (Online version in colour.)

comparatively older than those in Central Panama (figure 3, electronic supplementary material, figure S4). As discussed in the Material and methods, we do not assume the specific inferred dates are accurate given the likelihood that dynamics of *Bd* genome evolution violate several model assumptions. Our root-to-tip regression showed somewhat low temporal signal in our data ($R^2 = 0.02$) (electronic supplementary material, figure S5). However, our date randomization tests showed no overlap between real and randomized datasets, indicating a sufficient level of the temporal signal (electronic supplementary material, figure S6). While these results may appear contradictory, root-to-tip regression is a conservative approach assuming a strict molecular clock [47], while date randomization provides a more statistically robust method of comparison [48]. Therefore, as discussed in the Methods section above, our time-dated approaches are appropriate for inferring relative invasion histories across regions rather than proposing specific divergence or invasion dates.

For BEAST2, the time to most recent common ancestor (tMRCA) for Sierra Nevada samples was estimated to be 474 years from present day (95%HPD 510–393 years from present day) and estimated tMRCA in Central Panama was 277 years from present day (95%HPD 389–60 years from present day) (figure 3). For Nextstrain, tMRCA for Sierra Nevada samples was estimated as 1407 years from present day (95% CI 4498–1151) and tMRCA for Central Panama was estimated as 666 years from present day (95% CI 1914–534) (electronic supplementary material, figure S4); dynamic Nextstrain visualizations are available at: https://nextstrain.org/community/andrew-rothstein/bd-gpl/auspice/viz. Therefore,

even without ascribing weight to specific inferred dates, *Bd* in the Sierra Nevada appears to be much older than *Bd* in Panama. Confidence intervals for the inferred tMRCA do not overlap between regions with either analysis. The BEAST2 and Nexstrain time-dated phylogenetic approaches also corroborated PCA results (figure 1). Sierra Nevada samples largely clustered by the site while Central Panama samples had little to no structure based on-site location. (electronic supplementary material, figure S4) Finally, phylogenetic trees show an expected split within *Bd*GPL, supported by high BEAST posterior node values (electronic supplementary material, figure S7). This correspond to a previously reported split separating *Bd*GPL into two subclades: *Bd*GPL-1 and *Bd*GPL-2 [20,21]. Only GPL-2 is represented in Panama samples while GPL-1 and GPL-2 are both found in the Sierra Nevada samples.

## 4. Discussion

*Bd* has caused mass amphibian declines in many regions of the world and has been an example of the devasting effects of emerging wildlife diseases [12,15,16,35,52–54]. However, assessments of *Bd* emergence and spread have yet to incorporate genetically informed epizoology to examine pathogen dynamics at fine spatial scales. Our study used comparative population genetics to examine the genetic signatures of *Bd*GPL across two emblematic regions with disease-related amphibian declines. The alpine lakes of the Sierra Nevada and the tropical forests of Central Panama have dramatically different climate, habitat and host communities. However,

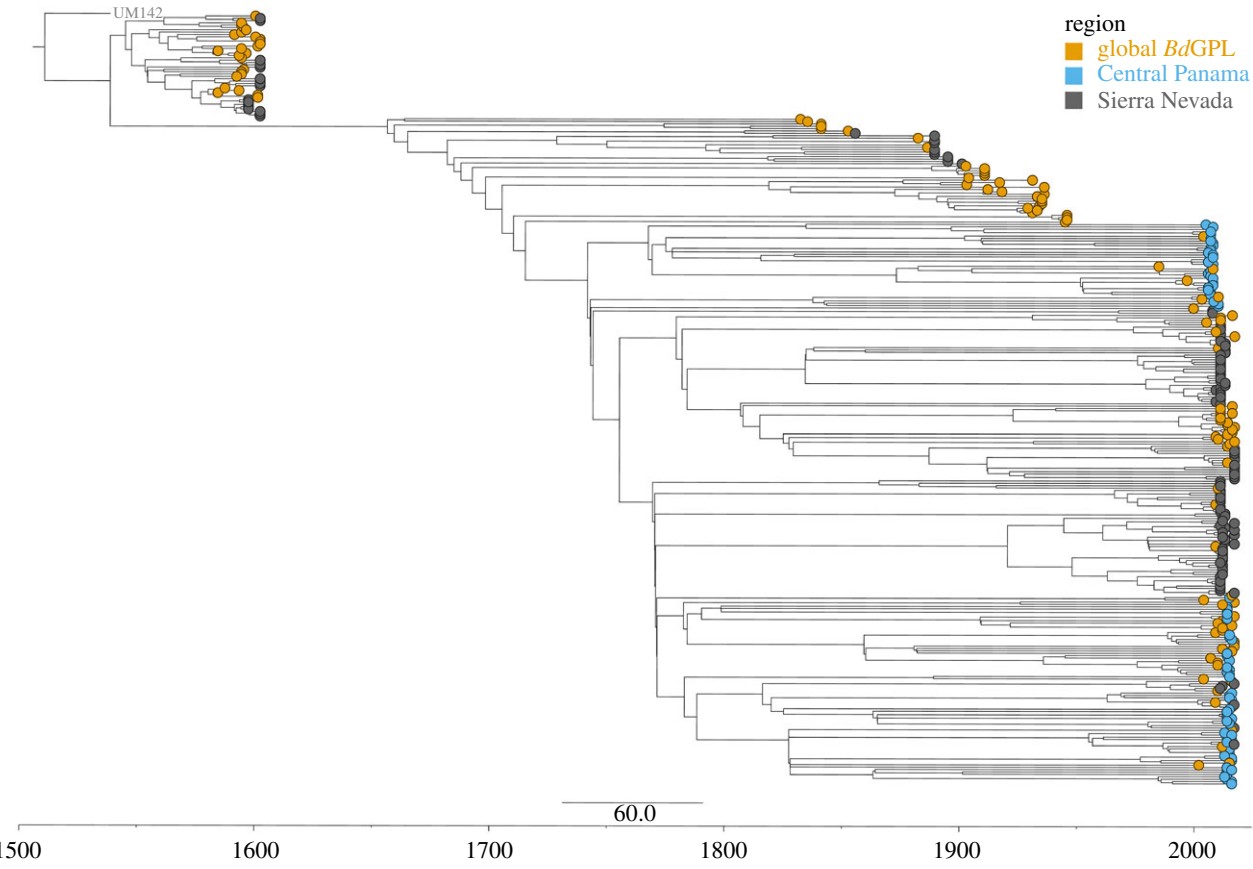

**Figure 3.** BEAST2 timed dated phylogeny among the Sierra Nevada, Central Panama, and global *Bd*GPL samples. Branch tips are colour coded by region. The tree is rooted by an outgroup from a more basal *Bd* lineage (*Bd*Brazil isolate UM142). Sierra Nevada samples are found across the tree, in multiple clusters, and with longer branch lengths than Central Panama samples suggesting a longer history of *Bd* in this region. (Online version in colour.)

they have been described as having similar histories of recent *Bd* emergence and spread. We tested the assumption that *Bd*GPL was recently introduced to these two regions and swept through each in a unidirectional epizootic wave. We found dramatic differences in *Bd* evolutionary history across regions, with an unexpectedly deep history of *Bd* in the Sierra Nevada. Here, we explore differences across regions providing a new perspective on these important historic declines. As wildlife disease rapidly continue to spread across the world, our framework is broadly applicable to interrogating observed patterns of pathogen emergence and spread to uncover important evolutionary pathogen histories.

## (a) How do patterns of pathogen genetic variation differ across regions?

### (i) *Bd*GPL in Central Panama is genetically similar and spatially unstructured

Our results from Central Panama support the hypothesis of a recent introduction, with *Bd* in this region lacking any spatial structure. All *Bd* genotypes from the Central Panama group tightly together, are generally distinct from *Bd* collected in the Sierra Nevada, and are all part of the GPL-2 subclade. This pattern supports previous studies reporting a single fast-moving outbreak of *Bd* through Central Panama [35]. Our samples from Central Panama were collected approximately 8 years after observed outbreaks (between 2012 and 2016), and the observed lack of genetic structure indicates that *Bd* did not diverge on a site-specific basis over this time period. Our findings support other recent studies

showing a lack of genetic, phenotypic and functional shifts in Central Panama *Bd* across similar temporal scales [32]. *Bd*GPL appears to have arrived in Panama much more recently than in the Sierra Nevada, maintained low levels of genetic diversity, and, over the last two decades, currently has no detectable genetic sub-structure.

### (ii) *Bd*GPL in Sierra Nevada is genetically diverse and spatially structured

We observed a dramatically different pattern in the Sierra Nevada, where we found high levels of genetic variation between sampling sites and spatial structuring of *Bd* genotypes. Although *Bd* samples were collected across a similar spatial and temporal scale as those from Panama, our genetic data indicate that *Bd*GPL has likely had a much longer historical presence in the Sierra Nevada than it has in Panama. This conclusion is supported by multiple lines of evidence. First, Sierra Nevada *Bd* contains more genetic variation and diversity than Central Panama (figure 2*a*). Measures of nucleotide diversity ($\pi$), are higher in Sierra Nevada *Bd* samples compared to Central Panama and Sierra Nevada *Bd* genetic diversity (Watterson's $\theta$) is significantly higher than the entire global panel of *Bd*GPL samples (figure 2*b*). This result is consistent with previous evidence that *Bd*GPL in the Sierra Nevada has higher levels of genetic diversity than *Bd*GPL from Arizona, Mexico, or Central Panama [55]. Second, we also observed a surprising pattern of spatially structured genetic diversity for *Bd*GPL in the Sierra Nevada. Sierra Nevada *Bd*GPL genotypes typically cluster by site and segregate by geographical distance in PC space

and in the phylogeny (figures 1*b* and 3*b*). Much of the observed genetic structure in the Sierra Nevada is consistent with a pattern of isolation by distance, suggesting a much longer history of *Bd* on the landscape. Third, even the exceptions to the pattern of isolation by distance suggests a deeper and more complex history of *Bd* in the Sierra Nevada. Samples from LeConte Divide and Conness Pond are genetically distinct from all other samples in the Sierra Nevada and cluster in PC space (figure 1*b*). These samples belong to a separate, early branching clade referred to as GPL-1 (figure 3). The presence of both *Bd*GPL-1 and *Bd*GPL-2 subclades could represent multiple independent introductions or much deeper *in situ* divergence, possibilities we revisit below.

## (b) What do regional differences suggest about *Bd*GPL origin and invasion history?

### (i) *Bd*GPL in Sierra Nevada likely predates the most recently observed wave of declines

One key factor that could contribute to radically different patterns of *Bd* genetic variation between Central Panama and the Sierra Nevada is invasion history (the timing and number of introductions). Our Nextstrain and BEAST2 analyses infer that *Bd* from the Sierra Nevada is older than *Bd* from Central Panama (figure 3, electronic supplementary material, figure S4). While our inference indicates that *Bd*GPL has been in the Sierra Nevada longer than Central Panama, it is difficult to assert specific invasion dates. As discussed in the Material and methods section, patterns of *Bd* genome evolution may violate a number of model assumptions. Although our analyses used a species-specific mutation rate inferred from *Bd* whole-genome analyses [6] our assay targets regions of the *Bd* genome that are most informative for discriminating among *Bd* lineages [42] and therefore may not evolve with a shared background mutation rate. Even without specific introduction dates, studies using histology and qPCR to test for *Bd* presence in museum specimens have often shown *Bd* presence prior to field-observed die-offs [24,56,57], which could indicate older introduction timings than previously assumed. As such, *Bd* presence has been detected in samples as far back as 1932 in the Sierra Nevada [24] and 1964 in Costa Rica (adjacent to Panama) [56].

Moreover, field observations suggest that *Bd* may be present in the environment well before an outbreak is observed. In some lakes, *Bd* is present at almost undetectably low prevalence and load for years before *Bd* loads spike and die-offs occur [16,28,58]. In some systems, *Bd* can even be detected from eDNA surveys before die-offs occur [59]. Such dynamics challenge our *a priori* expectations that *Bd* die-offs occur immediately after the pathogen first arrives in an area. In some systems, such as the Sierra Nevada and parts of Costa Rica [24,56,57], it is possible that *Bd* had a more wide-spread presence earlier than perceived. Whether there actually were earlier *Bd-caused* die-offs remains an open question. Increased surveillance of *Bd* before and during early outbreaks is needed to decouple initial pathogen invasion from observed pathogen-induced declines.

### (ii) The Sierra Nevada is a potential source for *Bd*GPL

High levels of genetic variation, deep spatial genetic structure and the presence of both subclades of *Bd*GPL in the Sierra Nevada suggest a longer evolutionary history of *Bd* in the

region than previously appreciated. The presence of both *Bd*GPL-1 and *Bd*GPL-2 could represent multiple asynchronous invasions of *Bd*GPL, a hypothesis raised by another recent spatial–temporal study of *Bd* presence in the Sierra Nevada [24]. An alternative explanation is that California is a potential *source* of *Bd* that has spread to other regions. As sampling resolution improves, it is possible that we will find other regions of the world with highly diverse and spatially structured *Bd*GPL populations. However, it is also worth continuing to challenge our assumptions about the origin and spread of this lineage. While the most basal lineage of *Bd* is from Asia [6], the origin of *Bd*GPL remains highly uncertain. Although we often assume that *Bd*GPL presence results from recent invasions, the region from which *Bd*GPL originated would be expected to have general characteristics similar to what we observe in the Sierra Nevada (i.e. relatively high genetic diversity and deep spatial structure). No such region other than the Sierra Nevada has yet been identified. Global sampling with greater spatial and temporal resolution will be needed to ultimately determine the origins of this highly virulent *Bd* lineage.

## (c) How do biotic and abiotic factors influence observed *Bd* genetic variation?

### (i) Differences in topography, host life history, community structure and climate also likely contribute to divergent patterns of pathogen genetic structure across regions

Biotic and abiotic factors also likely influence patterns of *Bd* genetic variation in a consistent direction, with increased opportunity for pathogen mixing in Central Panama relative to the Sierra Nevada. Central Panama is home to a diverse amphibian assemblage, with dozens of sympatric species that use a variety of microhabitats and have different reproductive modes [15,31]. A diverse host community in Panama with year-round activity and some direct-developing species (i.e. those without an aquatic larval phase) could provide more opportunities for *Bd* spread [31,60]. Central Panama contains landscape features that may be barriers to dispersal for some amphibian species [61], but interconnected stream networks still allow for fairly high connectivity among sites. By contrast, in the Sierra Nevada, our samples are from the only common—and highly susceptible—amphibian species in the alpine lake habitats (*R. sierrae/muscosa*) [62]. *Rana sierrae/muscosa* have high site fidelity, limited overland movements, spend the majority of each year under ice and inhabit disjunct alpine lakes separated by high mountain passes [63]. These features all impede connectivity among host populations and provide fewer opportunities for *Bd* dispersal [64]. Therefore, landscape and host factors consistently provide decreased opportunities for *Bd* gene flow in the Sierra Nevada, which is reflected in the greater pathogen spatial structure in this region.

In addition, Central Panama is significantly warmer and wetter than the Sierra Nevada. Temperature differences are particularly important because warmer temperatures (to a point) can lead to faster pathogen growth, an increased number of generations per year and greater opportunity for rapid evolutionary change. Slower *Bd* growth, generation time and evolutionary rates in the Sierra Nevada compared to Central Panama, make the patterns of higher genetic

diversity and strong spatial genetic structure in the Sierra Nevada all the more interesting.

## (b) How can pathogen genetic data help inform wildlife disease mitigation efforts?

Ultimately, integrating genetic, spatial and epizootic data within an evolutionary framework is a powerful way to understand the dynamics of emerging diseases of wildlife. Typically, studies of wildlife disease dynamics rely on *a priori* assumptions about pathogen introductions (i.e. based on earliest infection known from wild populations or museum records). However, our results, using *Bd* as an example for global pathogens, clearly demonstrates that outbreaks with similar epizoological signatures can still have radically different underlying pathogen histories. In our study, two regions with similar observed epizoological patterns in the field exhibit dramatically different pathogen evolutionary histories. In fact, one of the regions—the Sierra Nevada—has considerable pathogen diversity and genetic structure. Supporting evidence suggests that *Bd* in this region may persist in populations of highly susceptible host species at very low levels over many years without causing epizootics, opening the possibility that the pathogen has a much longer evolutionary history than previously appreciated. When we treat all population declines as the same, we overlook important nuances that could assist on-the-ground recovery and mitigation efforts. For example, if we incorporate *Bd* genotype data into choices of donor frog populations when planning translocations and reintroductions, we can mitigate human-induced mixing of *Bd* genotypes. Such actions could be an important component for species recovery efforts. By combining genetic and epizoological data, we can better understand differences in pathogen invasion history across regions and support more effective policies for biodiversity conservation and management.

Ethics. All sample collections were authorized by research permits provided by NPS, USFWS and the Institutional Animal Care and Use Committees of UC Berkeley, UC Santa Barbara, University of Nevada, Reno and the University of Pittsburgh.

Data accessibility. Processed data including BEAST2 tree XML, VCF file, genetic diversity values, tests for temporal signal and code for figures: https://figshare.com/s/0b3bcabff81fae2fb8e8.
Raw sequences of new samples deposited in NCBI SRA BioProject ID PRJNA686993: https://www.ncbi.nlm.nih.gov/bioproject/PRJNA686993.
Nextstrain code: https://github.com/andrew-rothstein/bd-gpl.git. The data are provided in the electronic supplementary material [65].

Authors' contributions. A.P.R.: conceptualization, data curation, formal analysis, investigation, methodology, visualization, writing—original draft, writing—review and editing; A.Q.B.: conceptualization, data curation, formal analysis, investigation, methodology, visualization, writing—original draft, writing—review and editing; R.A.K.: conceptualization, data curation, funding acquisition, investigation, methodology, project administration, supervision, visualization, writing—original draft, writing—review and editing; C.J.B.: conceptualization, data curation, funding acquisition, investigation, project administration, supervision, writing—original draft, writing—review and editing; J.V.: conceptualization, data curation, funding acquisition, investigation, project administration, supervision, writing—original draft, writing—review and editing; C.L.R.: conceptualization, data curation, funding acquisition, investigation, project administration, supervision, writing—original draft, writing—review and editing; E.B.R.: conceptualization, data curation, funding acquisition, investigation, methodology, project administration, supervision, visualization, writing—original draft, writing—review and editing.
All authors gave final approval for publication and agreed to be held accountable for the work performed therein.

Competing interests. We declare we have no competing interests.

Funding. This work was supported by DoD SERDP contract RC-2638 (to E.B.R., C.L.R.Z., J.V., C.J.B.), NSF DEB-1557190 (to E.B.R., C.J.B., R.A.K.), NSF DEB-1551488 (to E.B.R., C.L.R.Z., J.V.), NSF DEB CAREER - 1846403 (to J.V.), NSF DEB-166311 (to C.L.R.Z.), and the National Park Service. Sequencing done at IBEST Genomics Resources Core at the University of Idaho was supported in part by NIH COBRE grant P30GM103324. All sample collections were authorized by research permits provided by NPS, USFWS and the Institutional Animal Care and Use Committees of UC Berkeley, UC Santa Barbara, University of Nevada—Reno and University of Pittsburgh.

Acknowledgements. We thank the Sierra Nevada Aquatic Research Laboratory field crew, Danny Boiano from the National Park Service, Matt Robak, Angie Estrada, Renwei Chen and Mary Toothman for assisting in field collection and pathogen qPCR.

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
