## [Peer Review File · Proceedings of the Royal Society B: Biological Sciences]

Review History

RSPB-2020-3176.R0 (Original submission)

Review form: Reviewer 1

Recommendation

Accept with minor revision (please list in comments)

Scientific importance: Is the manuscript an original and important contribution to its field?

Good

General interest: Is the paper of sufficient general interest?

Acceptable

Quality of the paper: Is the overall quality of the paper suitable?

Good

Is the length of the paper justified?

Yes

Should the paper be seen by a specialist statistical reviewer?

No

Do you have any concerns about statistical analyses in this paper? If so, please specify them explicitly in your report.

No

It is a condition of publication that authors make their supporting data, code and materials available - either as supplementary material or hosted in an external repository. Please rate, if applicable, the supporting data on the following criteria.

Is it accessible?

Yes

Is it clear?

Yes

Is it adequate?

Yes

Do you have any ethical concerns with this paper?

No

Comments to the Author

In their paper, the authors use genetic sequences of *Batrachochytrium dendrobatidis* (Bd) from skin swabs to test whether the population genetic history of this fungal pathogen varies across sites in two regions where widespread Bd-induced declines have been recorded. Their results suggest greater spatial structuring and a deeper history of the pathogen in the Sierra Nevadas compared to sites in Panama based on genetic clustering, genetic diversity scores, and a dated phylogeny of their sampled strains and additional genotypes from the global pandemic lineage of Bd.

Overall I have relatively few comments on the manuscript, the analysis is presented in a clear and straightforward manner and it is well-written. The manuscript reads fairly speculative in nature, but this seems inherent to this sort of investigation on the population genetics of a pathogen. My only main comment is whether the authors can be confident that certain Bd genotypes do not display fidelity to particular amphibian host species that could in turn override spatial patterns. This seems a potential explanation (and the authors do touch on this in the discussion) for why the Bd genotypes appear to exhibit no spatial structure in Panama – they may cluster by host species clade instead. To that end, I think it would be interesting to display the samples along the principal component axes by species (or higher taxonomic rank) identities. I note that in their visualization of the PCA, there appears to be some clustering assignment of the sites, but I did not see the basis of this is in the methods. Similarly in Figure 2A, where there are not assigned clusters, how do the sites from the visualizations in Figure 1 match onto this broader sample? It would be interesting to see whether the samples in the second pandemic lineage all come from the same sites.

Minor comments

Line 77: in "the" late 1990s...

112: is "a" recent invasive pathogen

146-147: The number of different species in each region should also be listed here.

Review form: Reviewer 2

Recommendation

Major revision is needed (please make suggestions in comments)

Scientific importance: Is the manuscript an original and important contribution to its field?

Good

General interest: Is the paper of sufficient general interest?

Good

Quality of the paper: Is the overall quality of the paper suitable?

Poor

Is the length of the paper justified?

No

Should the paper be seen by a specialist statistical reviewer?

No

Do you have any concerns about statistical analyses in this paper? If so, please specify them explicitly in your report.

No

It is a condition of publication that authors make their supporting data, code and materials available - either as supplementary material or hosted in an external repository. Please rate, if applicable, the supporting data on the following criteria.

Is it accessible?

No

Is it clear?

No

Is it adequate?

Yes

Do you have any ethical concerns with this paper?

No

Comments to the Author

Rothstein, et al. present a study of Bd disease demographics around the world, with a focus on populations in California and Panama. While the topic is a good one, the presented data are insufficient to support the study. The materials and methods presented are inadequate and do not describe the underlying data, how it was produced, or its analysis sufficiently. Their results, including a principle components analysis, a set of SNP counts (Watterson's theta and Pi), and a tree, seem to present ambiguous results, despite the authors advertising a highly detailed analysis in the abstract and introduction based on a large collection of loci. While the PCA plot showing no structure in Panamanian populations is convincing, the plot depicting California populations and the inclusion of global populations is not (Fig 2). While the nucleotide statistics seem to have generated some significant differences, visually, the plots are not particularly convincing. This is confounded by the fact that one of the datasets is "global" Bd populations, which must include a huge and varying amount of geographically conflicting samples when compared to California and Panama. Given this data, the authors use ambiguous language like, "We confirmed that Bd from Sierra Nevada and Central Panama belong to the global BdGPL

lineage” without any explanation as to what that means with regards to their specific data (genotyped population loci). The abstract highlights their use of time-dated phylogenies, however, in the Results they state that their time dating is not accurate (but they continue to use it with no further justification). The tree they produce (Fig 3, in the main manuscript) seems to have a number of mixed nodes, containing global samples, California samples, and even the Panamanian samples are split in two clades, but this is not discussed in detail. The manuscript does mention GPL subclades (GPL-1 and GPL-2) but these are not labeled on the figures so it is hard to understand how these factors all interact. Compared with the three paragraphs of results, the authors present four pages of discussion that don’t (to this reviewer) seem well anchored in their empirical results.

In summary, the authors need to provide much more clarity on their methodological details, as well as their sampling details. For example, I was never fully able to understand the difference between the samples they collected in California and Panama versus the global Bd samples. The results need to be more specific and detailed, and the discussion needs to be scaled back to a large degree to specifically discuss what their results indicated. I think it would be useful to include more of the geographical data from the global data set (e.g. how do samples from Africa vs. Asia fall in the phylogenetic trees or PCA plots) in the analysis, as well as bringing the host species that were looked at (there were two in California, but many more in Panama) and discussing how host does (or does not) affect the analysis and filtering, etc.

Details are provided below.

Abstract

Line 51-52: you cite “time-dated phylogenies” in the abstract as a major data point, however your manuscript downplays the accuracy of these time measures at multiple points, this language needs to reflect your actual findings.

Materials and Methods

Line 142-143: How many of each species of host frog species were sampled?

Line 144-146: What type of data are the global BdGPL samples? You have classified them by continent, so are these sequenced data from single individuals, or are there populations that they are drawn from (yielding allele frequencies)? Were these biological samples that you sequenced or were they already processed and in a public database?

Line 147: What does it mean to say you “pre-amplified DNA”? What method is actually used here? PCR? Whole genome amplification?

Line 148: What are the 240 regions that you extracted? How did you choose those regions and why? Was it done using another reference genome, or are these randomized regions resulting from some type of restriction enzyme treatment, etc.?

Line 149: Absolutely unacceptable to list “similar methods to a custom assay” as a description of your analysis.

Line 149: “Post-sequencing...” How were the data sequenced? On what machine? How were they multiplexed into libraries, etc. All of this is critical data that needs to be included here.

Line 150-151: How were “ambiguity sequences” created? What process determined there was more than one nucleotide present?

Did you have both types of data for all samples, including the global BdGPL samples? If not, what other types of data did you integrate?

Line 153: How were “alignment, variant calling, and filtering” performed? The reader needs at least a description in the main text.

Most of this detail is in the supplement but should be here in the main manuscript. It is essential for understanding the underlying data and resulting analysis.

Besides categorizing which host species the Bd samples came from, you need to list the number of loci/SNPs that you filtered (providing the final number is not sufficient) to judge the overall reliability of the analysis.

Line 156: It is still unclear how the BdGPL samples fit into the dataset. Were they sequenced/processed in the same way as the Cali/Panama samples?

Line 159: “Given that sample sizes can greatly impact diversity metrics,” how do sample sizes impact diversity metrics? When you randomly sampled, how did it affect which samples you ended up keeping from California? Did you keep all sampling sites and all species hosts? Or did you drop some? A better approach would be to complete the analysis with and without subsampling and compare, is it necessary to subsample? (Most population genetic statistics are normalized to sample size.)

Line 161: 49 amplicons are removed from what, the 240 original amplicons or the 235 amplicons after filtering? Why were they removed?

Line 171: “We removed amplicons that had no data and included samples that had least 20 amplicons.” This is an additional set of filtering criteria. How are these filters different from those you applied above, why choose another set of filters? Surely if a sample has 20 or less amplicons it is not useful for the genetic diversity analysis? Why use MUSCLE to align and what are you aligning against? Didn’t you already implicitly align when you did the BWA alignment to a reference genome?

Line 177-178: How did you come up with this mutation rate? How did you generate the extended Bayesian Skyline plot, and what does it represent about your data set from a perspective of demographics?

Line 183: I believe the software is called “FigTree.”

Results

Figure 1A: It would be very helpful if the populations were labeled on the geographical maps so it is easier to compare visually with the PC clusters.

Figure 1B: For example, I see a yellow circle on the geographical map in Fig 1A, but can I assume this is “Marjorie Basin”, in Fig 1B. If so, why is there no yellow ellipse in the PCA plot?

Line 199: “We confirmed that Bd from Sierra Nevada and Central Panama belong to the global BdGPL lineage.” What does this mean, specifically? What data are the authors referring to and what does it mean to confirm?

Lines 212-215: “As discussed in the Material and methods, we do not assume the specific inferred dates are accurate.” I did not find any discussion of accuracy in the Materials and Methods, how the mutation rate was determined was not described (see my earlier comment).

Review form: Reviewer 3

Recommendation

Accept with minor revision (please list in comments)

Scientific importance: Is the manuscript an original and important contribution to its field?

Excellent

General interest: Is the paper of sufficient general interest?

Good

Quality of the paper: Is the overall quality of the paper suitable?

Excellent

Is the length of the paper justified?

Yes

Should the paper be seen by a specialist statistical reviewer?

No

Do you have any concerns about statistical analyses in this paper? If so, please specify them explicitly in your report.

Yes

It is a condition of publication that authors make their supporting data, code and materials available - either as supplementary material or hosted in an external repository. Please rate, if applicable, the supporting data on the following criteria.

Is it accessible?

Yes

Is it clear?

Yes

Is it adequate?

Yes

Do you have any ethical concerns with this paper?

No

Comments to the Author

The authors present a genetic study of *Batrachochytrium dendrobatidis* (Bd) from two regions: Central Panama and the Californian Sierra Nevada. Despite the prevailing hypothesis that the Global Panzootic Lineage of Bd (BdGPL) was recently introduced to both regions and spread quickly in a wave-like manner, the authors found that BdGPL has an evolutionary history that differed substantially between the regions. BdGPL in Central Panama conformed to the prevailing hypothesis, being genetically homogeneous and indicative of a more recent and rapid spread of BdGPL in this region. However, a high degree of spatial structuring was observed in Sierra Nevada populations to a degree that has not previously been observed anywhere in the world, as well as a much older phylogenetic history than was previously assumed.

Through the use of new sequencing approaches to achieve a significantly greater sampling density than has been feasible previously, this paper provides a clearer picture of fine-scale evolutionary relationships in a pathogen of global conservation concern. In doing so, the authors are able to challenge widely-held views about the dynamics of Bd spread and its impacts on

amphibian populations. This study is well-designed, explained, and reasoned. I found the writing to be clear, concise, and of an excellent standard overall. Despite some relatively minor concerns (see below), the methods were generally sound, and I believe the authors' interpretations were thoughtful and insightful and did not "over-sell" the results.

Although the introduction was generally strong, one thing that could be enhanced is the motivation for this particular study: specifically, the comparison between the Sierra Nevada and Central Panama. The authors do mention at line 114 that the two regions differ in climate, habitat, and community structure, but I think a stronger case could be made for why these factors might drive differential evolutionary histories between the two. There is a great deal of literature describing how Bd prevalence, transmission, and population impacts can vary depending on temperature and habitat. This literature establishes quite a strong precedent for finding differences in evolutionary history between regions but this at present is barely touched on by the authors.

My remaining comments largely pertain to the phylogenetic analysis. The authors state at line 177 that they used the GTR substitution model in BEAST2. Were any other models (e.g., HKY) also tested? If not, the authors should make use of BEAST2's features for estimating marginal likelihoods (e.g., path/stepping stone sampling) to test relative support for different models, or provide some other justification for their choice of model. In addition, which molecular clock (e.g., strict, uncorrelated lognormal relaxed) was used? Relative support for different clocks can and should also be evaluated in this way.

Although the authors address the potential limitations of their dataset for estimating specific dates, I would like to see this dealt with more quantitatively. Methods such as root-to-tip regression or the date randomization test can be used to test for temporal signal in genetic datasets and thus their suitability for inference of evolutionary timescales. Inclusion of such a test would enable easier interpretation of temporal results.

Finally, I would like to see node support indicated on the phylogeny (Figure 3). Posterior probabilities could be shown as numerical values or depicted using icons or colors. Ideally, this would be included in the main figure, but if adding this information to such a large tree compromises readability, a supplementary figure could be used to depict node support instead.

Typically, I would provide a list of brief minor comments that point out issues of clarity and structure in the text. However, the quality of writing was such that I did not identify any. Instead, I found only a few typographical errors, so I will point those out here:

Line 95: Should be "mountain yellow-legged frogs"

Line 109: Missing period after the citation.

Line 112: Missing "a" after "is".

Lines 182-183: Please provide citations for TreeAnnotator and FigureTree.

Line 257: Drop the "s" from "supports"

Decision letter (RSPB-2020-3176.R0)

16-Feb-2021

Dear Dr Rothstein:

I am writing to inform you that your manuscript RSPB-2020-3176 entitled "Divergent regional evolutionary histories of a devastating global amphibian pathogen" has, in its current form, been rejected for publication in Proceedings B.

This action has been taken on the advice of referees, who have recommended that substantial revisions are necessary. With this in mind we would be happy to consider a resubmission, provided the comments of the referees are fully addressed. However please note that this is not a provisional acceptance.

Sincerely,
Professor Hans Heesterbeek
mailto: proceedingsb@royalsociety.org

Associate Editor
Comments to Author:

The manuscript has been evaluated by three expert reviewers and by me. There is a consensus among us that there is considerable merit in this comparative study and that the findings could be of broad interest for the insight they provide into the global emergence and geographic spread of Bd. Several reviewers raised questions concerning host species identity and how this contributes to apparent patterns in the data. Reviewers also raised questions concerning the statistical and genetic methods used and presentation of the data. At least one reviewer could not access the NCBI entry. In considering a resubmission of their manuscript, I encourage the authors to conduct a detailed revision in response to the points raised by the reviewers.

Board Member: 2
Comments to Author(s):

In their manuscript entitled "Divergent regional evolutionary histories of a devastating global amphibian pathogen", Rothstein et al. utilize preserved Bd samples from amphibian populations in two regions that have experienced significant amphibian declines to conduct a rigorous genomic analysis of the evolutionary histories of these two distinct epizootics of chytridiomycosis. Interestingly, their analysis suggests different histories of invasion by Bd in

these two regions and potentially different patterns of geographic spread, challenging current understanding of patterns of Bd invasion in novel host communities.

Reviewer(s)' Comments to Author:

Referee: 1

Comments to the Author(s)

In their paper, the authors use genetic sequences of *Batrachochytrium dendrobatidis* (Bd) from skin swabs to test whether the population genetic history of this fungal pathogen varies across sites in two regions where widespread Bd-induced declines have been recorded. Their results suggest greater spatial structuring and a deeper history of the pathogen in the Sierra Nevadas compared to sites in Panama based on genetic clustering, genetic diversity scores, and a dated phylogeny of their sampled strains and additional genotypes from the global pandemic lineage of Bd.

Overall I have relatively few comments on the manuscript, the analysis is presented in a clear and straightforward manner and it is well-written. The manuscript reads fairly speculative in nature, but this seems inherent to this sort of investigation on the population genetics of a pathogen.

My only main comment is whether the authors can be confident that certain Bd genotypes do not display fidelity to particular amphibian host species that could in turn override spatial patterns. This seems a potential explanation (and the authors do touch on this in the discussion) for why the Bd genotypes appear to exhibit no spatial structure in Panama – they may cluster by host species clade instead. To that end, I think it would be interesting to display the samples along the principal component axes by species (or higher taxonomic rank) identities. I note that in their visualization of the PCA, there appears to be some clustering assignment of the sites, but I did not see the basis of this in the methods. Similarly in Figure 2A, where there are not assigned clusters, how do the sites from the visualizations in Figure 1 match onto this broader sample? It would be interesting to see whether the samples in the second pandemic lineage all come from the same sites.

Minor comments

Line 77: in "the" late 1990s...

112: is "a" recent invasive pathogen

146-147: The number of different species in each region should also be listed here.

Referee: 2

Comments to the Author(s)

Rothstein, et al. present a study of Bd disease demographics around the world, with a focus on populations in California and Panama. While the topic is a good one, the presented data are insufficient to support the study. The materials and methods presented are inadequate and do not describe the underlying data, how it was produced, or its analysis sufficiently. Their results, including a principle components analysis, a set of SNP counts (Watterson's θ and π), and a tree, seem to present ambiguous results, despite the authors advertising a highly detailed analysis in the abstract and introduction based on a large collection of loci. While the PCA plot showing no structure in Panamanian populations is convincing, the plot depicting California populations and the inclusion of global populations is not (Fig 2). While the nucleotide statistics seem to have generated some significant differences, visually, the plots are not particularly convincing. This is confounded by the fact that one of the datasets is "global" Bd populations, which must include a huge and varying amount of geographically conflicting samples when compared to California and Panama. Given this data, the authors use ambiguous language like, "We confirmed that Bd from Sierra Nevada and Central Panama belong to the global BdGPL lineage" without any explanation as to what that means with regards to their specific data (genotyped population loci). The abstract highlights their use of time-dated phylogenies, however, in the Results they state that their time dating is not accurate (but they continue to use it with no further justification). The tree they produce (Fig 3, in the main manuscript) seems to have a number of mixed nodes, containing global samples, California samples, and even the

Panamanian samples are split in two clades, but this is not discussed in detail. The manuscript does mention GPL subclades (GPL-1 and GPL-2) but these are not labeled on the figures so it is hard to understand how these factors all interact. Compared with the three paragraphs of results, the authors present four pages of discussion that don't (to this reviewer) seem well anchored in their empirical results.

In summary, the authors need to provide much more clarity on their methodological details, as well as their sampling details. For example, I was never fully able to understand the difference between the samples they collected in California and Panama versus the global Bd samples. The results need to be more specific and detailed, and the discussion needs to be scaled back to a large degree to specifically discuss what their results indicated. I think it would be useful to include more of the geographical data from the global data set (e.g. how do samples from Africa vs. Asia fall in the phylogenetic trees or PCA plots) in the analysis, as well as bringing the host species that were looked at (there were two in California, but many more in Panama) and discussing how host does (or does not) affect the analysis and filtering, etc.

Details are provided below.

Abstract

Line 51-52: you cite "time-dated phylogenies" in the abstract as a major data point, however your manuscript downplays the accuracy of these time measures at multiple points, this language needs to reflect your actual findings.

Materials and Methods

Line 142-143: How many of each species of host frog species were sampled?

Line 144-146: What type of data are the global BdGPL samples? You have classified them by continent, so are these sequenced data from single individuals, or are there populations that they are drawn from (yielding allele frequencies)? Were these biological samples that you sequenced or were they already processed and in a public database?

Line 147: What does it mean to say you "pre-amplified DNA"? What method is actually used here? PCR? Whole genome amplification?

Line 148: What are the 240 regions that you extracted? How did you choose those regions and why? Was it done using another reference genome, or are these randomized regions resulting from some type of restriction enzyme treatment, etc.?

Line 149: Absolutely unacceptable to list "similar methods to a custom assay" as a description of your analysis.

Line 149: "Post-sequencing..." How were the data sequenced? On what machine? How were they multiplexed into libraries, etc. All of this is critical data that needs to be included here.

Line 150-151: How were "ambiguity sequences" created? What process determined there was more than one nucleotide present?

Did you have both types of data for all samples, including the global BdGPL samples? If not, what other types of data did you integrate?

Line 153: How were "alignment, variant calling, and filtering" performed? The reader needs at least a description in the main text.

Most of this detail is in the supplement but should be here in the main manuscript. It is essential for understanding the underlying data and resulting analysis.

Besides categorizing which host species the Bd samples came from, you need to list the number of loci/SNPs that you filtered (providing the final number is not sufficient) to judge the overall reliability of the analysis.

Line 156: It is still unclear how the BdGPL samples fit into the dataset. Were they sequenced/processed in the same way as the Cali/Panama samples?

Line 159: "Given that sample sizes can greatly impact diversity metrics," how do sample sizes impact diversity metrics? When you randomly sampled, how did it affect which samples you ended up keeping from California? Did you keep all sampling sites and all species hosts? Or did you drop some? A better approach would be to complete the analysis with and without subsampling and compare, is it necessary to subsample? (Most population genetic statistics are normalized to sample size.)

Line 161: 49 amplicons are removed from what, the 240 original amplicons or the 235 amplicons after filtering? Why were they removed?

Line 171: "We removed amplicons that had no data and included samples that had least 20 amplicons." This is an additional set of filtering criteria. How are these filters different from those you applied above, why choose another set of filters? Surely if a sample has 20 or less amplicons it is not useful for the genetic diversity analysis? Why use MUSCLE to align and what are you aligning against? Didn't you already implicitly align when you did the BWA alignment to a reference genome?

Line 177-178: How did you come up with this mutation rate? How did you generate the extended Bayesian Skyline plot, and what does it represent about your data set from a perspective of demographics?

Line 183: I believe the software is called "FigTree."

Results

Figure 1A: It would be very helpful if the populations were labeled on the geographical maps so it is easier to compare visually with the PC clusters.

Figure 1B: For example, I see a yellow circle on the geographical map in Fig 1A, but can I assume this is "Marjorie Basin", in Fig 1B. If so, why is there no yellow ellipse in the PCA plot?

Line 199: "We confirmed that Bd from Sierra Nevada and Central Panama belong to the global BdGPL lineage." What does this mean, specifically? What data are the authors referring to and what does it mean to confirm?

Lines 212-215: "As discussed in the Material and methods, we do not assume the specific inferred dates are accurate." I did not find any discussion of accuracy in the Materials and Methods, how the mutation rate was determined was not described (see my earlier comment).

Referee: 3

Comments to the Author(s)

The authors present a genetic study of *Batrachochytrium dendrobatidis* (Bd) from two regions: Central Panama and the Californian Sierra Nevada. Despite the prevailing hypothesis that the Global Panzootic Lineage of Bd (BdGPL) was recently introduced to both regions and spread quickly in a wave-like manner, the authors found that BdGPL has an evolutionary history that differed substantially between the regions. BdGPL in Central Panama conformed to the

prevailing hypothesis, being genetically homogeneous and indicative of a more recent and rapid spread of BdGPL in this region. However, a high degree of spatial structuring was observed in Sierra Nevada populations to a degree that has not previously been observed anywhere in the world, as well as a much older phylogenetic history than was previously assumed.

Through the use of new sequencing approaches to achieve a significantly greater sampling density than has been feasible previously, this paper provides a clearer picture of fine-scale evolutionary relationships in a pathogen of global conservation concern. In doing so, the authors are able to challenge widely-held views about the dynamics of Bd spread and its impacts on amphibian populations. This study is well-designed, explained, and reasoned. I found the writing to be clear, concise, and of an excellent standard overall. Despite some relatively minor concerns (see below), the methods were generally sound, and I believe the authors' interpretations were thoughtful and insightful and did not "over-sell" the results.

Although the introduction was generally strong, one thing that could be enhanced is the motivation for this particular study: specifically, the comparison between the Sierra Nevada and Central Panama. The authors do mention at line 114 that the two regions differ in climate, habitat, and community structure, but I think a stronger case could be made for why these factors might drive differential evolutionary histories between the two. There is a great deal of literature describing how Bd prevalence, transmission, and population impacts can vary depending on temperature and habitat. This literature establishes quite a strong precedent for finding differences in evolutionary history between regions but this at present is barely touched on by the authors.

My remaining comments largely pertain to the phylogenetic analysis. The authors state at line 177 that they used the GTR substitution model in BEAST2. Were any other models (e.g., HKY) also tested? If not, the authors should make use of BEAST2's features for estimating marginal likelihoods (e.g., path/stepping stone sampling) to test relative support for different models, or provide some other justification for their choice of model. In addition, which molecular clock (e.g., strict, uncorrelated lognormal relaxed) was used? Relative support for different clocks can and should also be evaluated in this way.

Although the authors address the potential limitations of their dataset for estimating specific dates, I would like to see this dealt with more quantitatively. Methods such as root-to-tip regression or the date randomization test can be used to test for temporal signal in genetic datasets and thus their suitability for inference of evolutionary timescales. Inclusion of such a test would enable easier interpretation of temporal results.

Finally, I would like to see node support indicated on the phylogeny (Figure 3). Posterior probabilities could be shown as numerical values or depicted using icons or colors. Ideally, this would be included in the main figure, but if adding this information to such a large tree compromises readability, a supplementary figure could be used to depict node support instead.

Typically, I would provide a list of brief minor comments that point out issues of clarity and structure in the text. However, the quality of writing was such that I did not identify any. Instead, I found only a few typographical errors, so I will point those out here:

Line 95: Should be "mountain yellow-legged frogs"

Line 109: Missing period after the citation.

Line 112: Missing "a" after "is".

Lines 182-183: Please provide citations for TreeAnnotator and FigureTree.

Line 257: Drop the "s" from "supports"

Author's Response to Decision Letter for (RSPB-2020-3176.R0)

See Appendix A.

RSPB-2021-0782.R0

Review form: Reviewer 1

Recommendation

Accept as is

Scientific importance: Is the manuscript an original and important contribution to its field?

Good

General interest: Is the paper of sufficient general interest?

Good

Quality of the paper: Is the overall quality of the paper suitable?

Good

Is the length of the paper justified?

Yes

Should the paper be seen by a specialist statistical reviewer?

No

Do you have any concerns about statistical analyses in this paper? If so, please specify them explicitly in your report.

No

It is a condition of publication that authors make their supporting data, code and materials available - either as supplementary material or hosted in an external repository. Please rate, if applicable, the supporting data on the following criteria.

Is it accessible?

Yes

Is it clear?

Yes

Is it adequate?

Yes

Do you have any ethical concerns with this paper?

No

Comments to the Author

Thank you for considering my questions.

Review form: Reviewer 2

Recommendation

Accept with minor revision (please list in comments)

Scientific importance: Is the manuscript an original and important contribution to its field?

Good

General interest: Is the paper of sufficient general interest?

Good

Quality of the paper: Is the overall quality of the paper suitable?

Good

Is the length of the paper justified?

Yes

Should the paper be seen by a specialist statistical reviewer?

No

Do you have any concerns about statistical analyses in this paper? If so, please specify them explicitly in your report.

No

It is a condition of publication that authors make their supporting data, code and materials available - either as supplementary material or hosted in an external repository. Please rate, if applicable, the supporting data on the following criteria.

Is it accessible?

Yes

Is it clear?

Yes

Is it adequate?

Yes

Do you have any ethical concerns with this paper?

No

Comments to the Author

The revised manuscript is much improved and looks good. I have four minor points that should be addressed before publication.

Table S1 is missing.

Authors need to report the mean (+Stdev or Stderr) depth of coverage for all of the samples from which theta and pi statistics were eventually drawn. For at least the Central American and Sierra Nevada samples, these should be reported in the main manuscript methods.

Methods, line 168: authors need to clarify and state that SNPs were aligned and called with BWA/FreeBayes, filtered with VCFTools, and the results fed into ANGSD to estimate the SFS (assuming I understand the analysis properly). This is important as ANGSD does its own native SNP calling on low-coverage data, which is a significantly different method from the haplotype-based calls of FreeBayes.

Discussion, line 292: the authors should not report a time of arrival in Central America (~250 years), since the tMRCA-based coalescent tree could only report relative differences between CA and the Sierra Nevada samples and the Nextstrain analysis differs from this value quite a lot. Time can be reported in a relative amount here.

Line 299: Period missing at end of sentence.

Decision letter (RSPB-2021-0782.R0)

21-May-2021

Dear Dr Rothstein

I am pleased to inform you that your manuscript RSPB-2021-0782 entitled "Divergent regional evolutionary histories of a devastating global amphibian pathogen" has been accepted for publication in *Proceedings B*, subject to some final minor revision.

The referees have recommended publication, but one reviewer suggests some minor revisions to your manuscript. Therefore, I invite you to respond to the referee's comments and revise your manuscript. Because the schedule for publication is very tight, it is a condition of publication that you submit the revised version of your manuscript within 7 days. If you do not think you will be able to meet this date please let us know.

[http://datadryad.org/submit?journalID=RSPB&manu=\(Document not available\)](http://datadryad.org/submit?journalID=RSPB&manu=(Document%20not%20available)) which will take you to your unique entry in the Dryad repository. If you have already submitted your data to dryad you can make any necessary revisions to your dataset by following the above link. Please see <https://royalsociety.org/journals/ethics-policies/data-sharing-mining/> for more details.

Sincerely,

Professor Hans Heesterbeek

Associate Editor

Comments to Author:

The authors have performed an extensive revision of their manuscript in response to reviewer feedback and as a result it is greatly improved. Two of the original reviewers have reviewed the resubmission and agree that the manuscript overall is important and highly interesting. Please make sure to address the few additional reviewer comments provided on the resubmission.

Reviewer(s)' Comments to Author:

Referee: 1

Comments to the Author(s).

Thank you for considering my questions.

Referee: 2

Comments to the Author(s).

The revised manuscript is much improved and looks good. I have four minor points that should be addressed before publication.

Table S1 is missing.

Authors need to report the mean (+Stdev or Stderr) depth of coverage for all of the samples from which theta and pi statistics were eventually drawn. For at least the Central American and Sierra Nevada samples, these should be reported in the main manuscript methods.

Methods, line 168: authors need to clarify and state that SNPs were aligned and called with BWA/FreeBayes, filtered with VCFTools, and the results fed into ANGSD to estimate the SFS (assuming I understand the analysis properly). This is important as ANGSD does its own native SNP calling on low-coverage data, which is a significantly different method from the haplotype-based calls of FreeBayes.

Discussion, line 292: the authors should not report a time of arrival in Central America (~250 years), since the tMRCA-based coalescent tree could only report relative differences between CA and the Sierra Nevada samples and the Nextstrain analysis differs from this value quite a lot. Time can be reported in a relative amount here.

Line 299: Period missing at end of sentence.

Author's Response to Decision Letter for (RSPB-2021-0782.R0)

See Appendix B.

Decision letter (RSPB-2021-0782.R1)

28-May-2021

Dear Dr Rothstein

I am pleased to inform you that your manuscript entitled "Divergent regional evolutionary histories of a devastating global amphibian pathogen" has been accepted for publication in Proceedings B.

Data Accessibility section

Open Access

Paper charges

Sincerely,

Appendix A

Andrew P. Rothstein, M.Sc., Ph.D.
Department of Environmental Science,
Policy, & Management

54 Mulford Hall
Berkeley, CA 94720
P:206-295-9766
E:andrew.rothstein@berkeley.edu

April 6, 2021

Dear Editors,

Thank you for a constructive review of our manuscript entitled “*Divergent regional evolutionary histories of a devastating global amphibian pathogen*” [RSPB-2020-3176] as a submission for *Proceedings of Royal Society B*. We appreciate the time, effort, and helpful comments from the three reviewers and editors. Our study closely re-examines two of the most iconic disease-associated amphibian community declines ever documented. By comparing fine-scale population genetic patterns of the devastating pathogenic amphibian chytrid fungus, we uncover a surprising contrast between the two systems that challenges long-held beliefs about the history of disease emergence – and amphibian declines – globally.

Based on reviewer and editor comments, we have addressed the following primary concerns and several minor comments throughout. Reviewers agreed that the addition of some clarifying figures and language would aid in results interpretation. We added supplementary figures that further support our findings of site-specific spatial genetic structuring of Bd in the Sierra Nevada. Additionally, we added more details to our Methods and Results, and supplied clarifying supplementary figures that address reviewer concerns regarding temporal signal in our phylodynamic analyses. We also moved some details originally included in our Supplementary Materials into our Methods section to provide clarity on our methodologies. Lastly, we addressed many minor comments and ensured that our data accessibility meets *Proceedings of Royal Society B* requirements. Our responses are provided point by point in the bold font below.

Reviewer comments have strengthened our manuscript, and we expect that our findings will spur new hypotheses within the amphibian disease community and to provide a valuable framework for genetic epizootological approaches for other emerging wildlife diseases. Thank you for your continued consideration.

Sincerely,

Andrew P. Rothstein (on behalf of authors)

All line numbers refer to track changes version of manuscript

Referee: 1

Comments to the Author(s)

In their paper, the authors use genetic sequences of *Batrachochytrium dendrobatidis* (Bd) from skin swabs to test whether the population genetic history of this fungal pathogen varies across sites in two regions where widespread Bd-induced declines have been recorded. Their results suggest greater spatial structuring and a deeper history of the pathogen in the Sierra Nevadas compared to sites in Panama based on genetic clustering, genetic diversity scores, and a dated phylogeny of their sampled strains and additional genotypes from the global pandemic lineage of Bd.

Overall I have relatively few comments on the manuscript, the analysis is presented in a clear and straightforward manner and it is well-written. The manuscript reads fairly speculative in nature, but this seems inherent to this sort of investigation on the population genetics of a pathogen.

My only main comment is whether the authors can be confident that certain Bd genotypes do not display fidelity to particular amphibian host species that could in turn override spatial patterns. This seems a potential explanation (and the authors do touch on this in the discussion) for why the Bd genotypes appear to exhibit no spatial structure in Panama – they may cluster by host species clade instead. To that end, I think it would be interesting to display the samples along the principal component axes by species (or higher taxonomic rank) identities. I note that in their visualization of the PCA, there appears to be some clustering assignment of the sites, but I did not see the basis of this is in the methods. Similarly in Figure 2A, where there are not assigned clusters, how do the sites from the visualizations in Figure 1 match onto this broader sample? It would be interesting to see whether the samples in the second pandemic lineage all come from the same sites.

We appreciate the suggestion to further explore Bd genotypes and evaluate potential site-specific or species-specific genetic structure. To address this we, we have added:

- 1) A supplementary figure (Figure S1) as a replicated Figure 1 with each point identified by site.**
- 2) A supplementary figure (Figure S2) as a replicated PCA of Figure 2C, highlighting that Bd spatial patterns are not due to species differences in Central Panama. We include only Central Panama in this new figure because it contains a rich amphibian community (whereas the Sierra Nevada system only contains two sister species).**

These added figures help support our findings that Bd genotypes in the Sierra Nevada cluster by site and not by other underlying host structuring. Further, we show that Panama Bd genotypes do not cluster by species.

Minor comments

Line 77: in "the" late 1990s...

Typo corrected. [Line 74]

112: is "a" recent invasive pathogen

Typo corrected. [Line 109]

146-147: The number of different species in each region should also be listed here.

Added species for each region in line. [Lines 142-143; Line 146]

Referee: 2

Comments to the Author(s)

Rothstein, et al. present a study of Bd disease demographics around the world, with a focus on populations in California and Panama. While the topic is a good one, the presented data are insufficient to support the study. The materials and methods presented are inadequate and do not describe the underlying data, how it was produced, or its analysis sufficiently. Their results, including a principle components analysis, a set of SNP counts (Watterson's theta and Pi), and a tree, seem to present ambiguous results, despite the authors advertising a highly detailed analysis in the abstract and introduction based on a large collection of loci. While the PCA plot showing no structure in Panamanian populations is convincing, the plot depicting California populations and the inclusion of global populations is not (Fig 2). While the nucleotide statistics seem to have generated some significant differences, visually, the plots are not particularly convincing. This is confounded by the fact that one of the datasets is "global" Bd populations, which must include a huge and varying amount of geographically conflicting samples when compared to California and Panama. Given this data, the authors use ambiguous language like, "We confirmed that Bd from Sierra Nevada and Central Panama belong to the global BdGPL lineage" without any explanation as to what that means with regards to their specific data (genotyped population loci). The abstract highlights their use of time-dated phylogenies, however, in the Results they state that their time dating is not accurate (but they continue to use it with no further justification). The tree they produce (Fig 3, in the main manuscript) seems to have a number of mixed nodes, containing global samples, California samples, and even the Panamanian samples are split in two clades, but this is not discussed in detail. The manuscript does mention GPL subclades (GPL-1 and GPL-2) but these are not labeled on the figures so it is hard to understand how these factors all interact. Compared with the three paragraphs of results, the authors present four pages of discussion that don't (to this reviewer) seem well anchored in their empirical results.

In summary, the authors need to provide much more clarity on their methodological details, as well as their sampling details. For example, I was never fully able to understand the difference between the samples they collected in California and Panama versus the global Bd samples. The results need to be more specific and detailed, and the discussion needs to be scaled back to a large degree to specifically discuss what their results indicated. I think it would be useful to include more of the geographical data from the global data set (e.g. how do samples from Africa vs. Asia fall in the phylogenetic trees or PCA plots) in the analysis, as well as bringing the host species that were looked at (there were two in California, but many more in Panama) and discussing how host does (or does not) affect the analysis and filtering, etc.

We appreciate Reviewer 2's comments, especially the need to add clarifying language to our methodology. We pulled language from our Supplementary Materials and added it to the main text to provide additional details in the Methods section. Specifically, we explain how incorporating the previously published global dataset of Bd genotypes helps contextualize our newly sequenced samples from Sierra Nevada and Central Panama. With regard to our time-dated phylogenies, we have added additional language originally in the Supplementary Materials to the main text detailing the caveats associated with time-dated approaches and added labels for GPL subclades to Figure 3. Below are responses to individual comments.

Details are provided below.

Abstract

Line 51-52: you cite "time-dated phylogenies" in the abstract as a major data point, however your manuscript downplays the accuracy of these time measures at multiple points, this language needs to reflect your actual findings.

We added language explaining the relative nature of our analytical methods. [Line 50]

Materials and Methods

Line 142-143: How many of each species of host frog species were sampled?

Edited as suggested from both Reviewer 1 and 2. For global species we identified total number of species and referred to published data in our Supplementary Table [Lines 142-143; Line 146]

Line 144-146: What type of data are the global BdGPL samples? You have classified them by continent, so are these sequenced data from single individuals, or are there populations that they are drawn from (yielding allele frequencies)? Were these biological samples that you sequenced or were they already processed and in a public database?

We added language describing the previously published global BdGPL samples and their context for our study. [Lines 145-149]

Line 147: What does it mean to say you “pre-amplified DNA”? What method is actually used here? PCR? Whole genome amplification?

We added an explanation of preamplification, pulled from supplementary materials. [Lines 151-153]

Line 148: What are the 240 regions that you extracted? How did you choose those regions and why? Was it done using another reference genome, or are these randomized regions resulting from some type of restriction enzyme treatment, etc.?

We added citation to our published methods about the 240 focal regions across the Bd genome. [Lines 152-153]

Line 149: Absolutely unacceptable to list “similar methods to a custom assay” as a description of your analysis.

We added a citation for our published methods paper, which used identical sample preparation and sequencing methods as this study. [Line 153]

Line 149: “Post-sequencing...” How were the data sequenced? On what machine? How were they multiplexed into libraries, etc. All of this is critical data that needs to be included here.

We added language, pulled from supplementary materials, to explain microfluidic PCR and sequencing. [Lines 152-164]

Line 150-151: How were “ambiguity sequences” created? What process determined there was more than one nucleotide present?

We added language for post-sequencing processing of samples including details on ambiguity sequences and raw fastq. [Lines 156-162]

Did you have both types of data for all samples, including the global BdGPL samples? If not, what other types of data did you integrate?

We added language, pulled from supplementary materials, explaining which types of data were used for different downstream applications. [Lines 161-164]

Line 153: How were “alignment, variant calling, and filtering” performed? The reader needs at least a description in the main text.

We clarified which parts of our methods are in the Supplementary Materials and per page limits referred reader to specific parameters used in alignment and variant calling. [Lines 151-164]

Most of this detail is in the supplement but should be here in the main manuscript. It is essential for understanding the underlying data and resulting analysis.

Besides categorizing which host species the Bd samples came from, you need to list the number of loci/SNPs that you filtered (providing the final number is not sufficient) to judge the overall reliability of the analysis.

We added the number of prefiltered variants as well as post filtering variants (with filtering steps in supplementary materials). [Lines 164]

Line 156: It is still unclear how the BdGPL samples fit into the dataset. Were they sequenced/processed in the same way as the Cali/Panama samples?

We added clarification about the global BdGPL samples with information about their sequence/processing [Lines 145-149]

Line 159: “Given that sample sizes can greatly impact diversity metrics,” how do sample sizes impact diversity metrics? When you randomly sampled, how did it affect which samples you ended up keeping from California? Did you keep all sampling sites and all species hosts? Or did you drop some? A better approach would be to complete the analysis with and without subsampling and compare, is it necessary to subsample? (Most population genetic statistics are normalized to sample size.)

The main diversity statistics across regions are presented in Figure 2B and 2C. When using ANGSD diversity statistics, inferences can be sensitive to sample size. Therefore, we 1) evened our sampling across regions by randomly choosing samples within region and 2) tested between Sierra Nevada and Central Panama sites to confirm that any differences in sample sizes among sites were not responsible for observed statistical differences between regions.

Line 161: 49 amplicons are removed from what, the 240 original amplicons or the 235 amplicons after filtering? Why were they removed?

Clarified as suggested. [Line 171-172]

Line 171: “We removed amplicons that had no data and included samples that had least 20 amplicons.” This is an additional set of filtering criteria. How are these filters different from those you applied above, why choose another set of filters? Surely if a sample has 20 or less amplicons it is not useful for the genetic diversity analysis? Why use MUSCLE to align and what are you aligning against? Didn’t you already implicitly align when you did the BWA alignment to a reference genome?

There were two types of data created from our sequences: raw fastq (by sample) and ambiguities (by amplicon and by sample). The raw fastq files were used for variant calling, PCA, and diversity statistics. The ambiguities were used for phylogenetics and time-dated analyses. Therefore, BWA alignments were applied to fastq and MUSCLE alignments were applied to ambiguities for each amplicon. Explanations are added in text. Specific details for methods in variant calling are in Supplementary Materials. [Lines 158-164]

Line 177-178: How did you come up with this mutation rate? How did you generate the extended Bayesian Skyline plot, and what does it represent about your data set from a perspective of demographics?

Both reviewer 2 and 3 asked for clarifying information on our phylogeny and tree model. We wanted our time dated phylogeny to be directly comparable to a recently published study of whole genome sequencing of Bd (O’Hanlon et al 2018 Science). Therefore, we used identical model parameters such as substitution model and clock rates. We have added language to be more explicit about this in our methods. [Lines 199-208]

Line 183: I believe the software is called “FigTree.”

Typo corrected. [Line 206]

Results

Figure 1A: It would be very helpful if the populations were labeled on the geographical maps so it is easier to compare visually with the PC clusters.

Figure 1B: For example, I see a yellow circle on the geographical map in Fig 1A, but can I assume this is “Marjorie Basin”, in Fig 1B. If so, why is there no yellow ellipse in the PCA plot?

While we acknowledge the reviewer’s comments about figure readability, adding additional labels instead of the color coordinate labelling scheme we used, we believe, would clutter the already large Figure 1. We have added language in our Figure 1 caption to cue the reader to the matching color scheme. With regard to Figure 1B, Marjorie Basin has a small sample size but is present in the figure. It is located within the Barrett Basin (light blue) ellipse. [Lines 616-617 and Figure 1]

Line 199: “We confirmed that Bd from Sierra Nevada and Central Panama belong to the global BdGPL lineage.” What does this mean, specifically? What data are the authors referring to and what does it mean to confirm?

Added clarifying language. See above comment on the inclusion of our BdGPL dataset. [Lines 239-240]

Lines 212-215: “As discussed in the Material and methods, we do not assume the specific inferred dates are accurate.” I did not find any discussion of accuracy in the Materials and Methods, how the mutation rate was determined was not described (see my earlier comment).

We added clarifying details, lifted from supplementary materials, of mutation rate including reference to whole genome work for time dated model. [Lines 196-224]

Referee: 3

Comments to the Author(s)

The authors present a genetic study of *Batrachochytrium dendrobatidis* (Bd) from two regions: Central Panama and the Californian Sierra Nevada. Despite the prevailing hypothesis that the Global Panzootic Lineage of Bd (BdGPL) was recently introduced to both regions and spread quickly in a wave-like manner, the authors found that BdGPL has an evolutionary history that differed substantially between the regions. BdGPL in Central Panama conformed to the prevailing hypothesis, being genetically homogeneous and indicative of a more recent and rapid spread of BdGPL in this region. However, a high degree of spatial structuring was observed in Sierra Nevada populations to a degree that has not previously been observed anywhere in the world, as well as a much older phylogenetic history than was previously assumed.

Through the use of new sequencing approaches to achieve a significantly greater sampling density than has been feasible previously, this paper provides a clearer picture of fine-scale evolutionary relationships in a pathogen of global conservation concern. In doing so, the authors are able to challenge widely-held views about the dynamics of Bd spread and its impacts on amphibian populations. This study is well-designed, explained, and reasoned. I found the writing to be clear, concise, and of an excellent standard overall. Despite some relatively minor concerns (see below), the methods were generally sound, and I believe the authors’ interpretations were thoughtful and insightful and did not “over-sell” the results.

Although the introduction was generally strong, one thing that could be enhanced is the motivation for this particular study: specifically, the comparison between the Sierra Nevada and Central Panama. The authors do mention at line 114 that the two regions differ in climate, habitat, and community structure, but I think a stronger case could be made for why these factors might drive differential evolutionary histories between the two. There is a great deal of literature describing how Bd prevalence, transmission, and population impacts can vary depending on temperature and habitat. This literature establishes quite a strong precedent for finding differences in evolutionary history between regions but this at present is barely touched on by the authors.

We have added additional language and citations, regarding environmental factors influencing evolutionary patterns in our introduction to improve clarity about our motivation for the study. [Lines 112-115]

My remaining comments largely pertain to the phylogenetic analysis. The authors state at line 177 that they used the GTR substitution model in BEAST2. Were any other models (e.g., HKY) also tested? If not, the authors should make use of

BEAST2's features for estimating marginal likelihoods (e.g., path/stepping stone sampling) to test relative support for different models, or provide some other justification for their choice of model. In addition, which molecular clock (e.g., strict, uncorrelated lognormal relaxed) was used? Relative support for different clocks can and should also be evaluated in this way.

This is a helpful comment pertaining to methodology clarifications. There are two reasons why we used our GTR substitution model. First, we used a best model approach built into the IQ-Tree program and found, through AIC scores, that GTR was the best fit model for our data. Second, we wanted our time dated phylogeny to be directly comparable to a recently published study of whole genome sequencing of Bd. Therefore, we used model parameters from a previous whole genome study for our study including substitution model and clock rates. We have added language to be more explicit about these points in our methods. [Lines 193-224]

Although the authors address the potential limitations of their dataset for estimating specific dates, I would like to see this dealt with more quantitatively. Methods such as root-to-tip regression or the date randomization test can be used to test for temporal signal in genetic datasets and thus their suitability for inference of evolutionary timescales. Inclusion of such a test would enable easier interpretation of temporal results.

We appreciate the reviewer's comments on limitations to our time-based phylogenies. While we caveat in our methods and discussion the limitations and assumptions of using time-dated phylogenies we have also included both language and supplementary figures that show we had low-moderate temporal signal in our data. We added methods of both a root-to-tip regression and a date randomization test to test for temporal signal. Both tests indicated we had sufficient temporal signal to proceed with our time-dated approaches. [Lines 193-224; Lines 259-265]

Finally, I would like to see node support indicated on the phylogeny (Figure 3). Posterior probabilities could be shown as numerical values or depicted using icons or colors. Ideally, this would be included in the main figure, but if adding this information to such a large tree compromises readability, a supplementary figure could be used to depict node support instead.

We agree with the reviewers that adding this information to a large tree would compromise readability, but we have added an additional supplemental figure with color coded branches based on posterior probabilities from our BEAST analyses (Figure S7).

Typically, I would provide a list of brief minor comments that point out issues of clarity and structure in the text. However, the quality of writing was such that I did not identify any. Instead, I found only a few typographical errors, so I will point those out here:

Line 95: Should be "mountain yellow-legged frogs"

Typo corrected. [Line 92]

Line 109: Missing period after the citation.

Typo corrected. [Line 106]

Line 112: Missing "a" after "is".

Typo corrected. [Line 109]

Lines 182-183: Please provide citations for TreeAnnotator and FigureTree.

Citations added for both software. [Line 206]

Line 257: Drop the "s" from "supports"

Typo corrected. [Line 312]

Appendix B

Andrew P. Rothstein, M.Sc., Ph.D.
Department of Environmental Science,
Policy, & Management

54 Mulford Hall
Berkeley, CA 94720
P:206-295-9766
E:andrew.rothstein@berkeley.edu

May 24th, 2021

Dear Editors,

Thank you for a constructive review and acceptance of our manuscript entitled “*Divergent regional evolutionary histories of a devastating global amphibian pathogen*” [RSPB-2020-3176] for *Proceedings of Royal Society B*. We appreciate the time, effort, and helpful comments from the two reviewers and editors. We agree that each review has strengthened our study. We expect that our findings will spur new hypotheses within the amphibian disease community and to provide a valuable framework for genetic epizootological approaches for other emerging wildlife diseases.

We have responded to reviewers’ comments and suggestions below. Thank you again.

Sincerely,

Andrew P. Rothstein (on behalf of authors)

Associate Editor

Comments to Author:

The authors have performed an extensive revision of their manuscript in response to reviewer feedback and as a result it is greatly improved. Two of the original reviewers have reviewed the resubmission and agree that the manuscript overall is important and highly interesting. Please make sure to address the few additional reviewer comments provided on the resubmission.

Reviewer(s)' Comments to Author:

Referee: 1

Comments to the Author(s).

Thank you for considering my questions.

Thank you for your review and strengthening our manuscript.

Referee: 2

Comments to the Author(s).

The revised manuscript is much improved and looks good. I have four minor points that should be addressed before publication.

Table S1 is missing.

We have re-uploaded Table S1 as a .csv file in electronic supplementary materials. We have also changed main text to refer reader to electronic supplementary material. [Line 146]

Authors need to report the mean (+Stdev or Stderr) depth of coverage for all of the samples from which theta and pi statistics were eventually drawn. For at least the Central American and Sierra Nevada samples, these should be reported in the main manuscript methods.

We have added mean amplicon depth with standard deviation for each region in our methods. [Lines 176-178]

Methods, line 168: authors need to clarify and state that SNPs were aligned and called with BWA/FreeBayes, filtered with VCFTools, and the results fed into ANGSD to estimate the SFS (assuming I understand the analysis properly). This is important as ANGSD does its own native SNP calling on low-coverage data, which is a significantly different method from the haplotype-based calls of FreeBayes.

We have clarified that ANGSD was used with filtered bam files to calculate diversity statistics – separate from our variant calling methods [Lines 169-170]

Discussion, line 292: the authors should not report a time of arrival in Central America (~250 years), since the tMRCA-based coalescent tree could only report relative differences between CA and the Sierra Nevada samples and the Nextstrain analysis differs from this value quite a lot. Time can be reported in a relative amount here.

We have removed language with specific dates for tMRCA and reiterated relative ages between regional samples. [Lines 295-297]

Line 299: Period missing at end of sentence.

Corrected as suggested. [Line 303]